# An intermetallic molecular nanomagnet with the lanthanide coordinated only by transition metals

Michał Magott [1], Maria Brzozowska [1], Stanisław Baran [2], Veacheslav Vieru [3] & Dawid Pinkowicz [1✉]

Magnetic molecules known as molecular nanomagnets (MNMs) may be the key to ultra-high density data storage. Thus, novel strategies on how to design MNMs are desirable. Here, inspired by the hexagonal structure of the hardest intermetallic magnet $SmCo_5$, we have synthesized a nanomagnetic molecule where the central lanthanide (Ln) $Er^{III}$ is coordinated solely by three transition metal ions (TM) in a perfectly trigonal planar fashion. This intermetallic molecule $[Er^{III}(Re^{I}Cp_2)_3]$ (**ErRe₃**) starts a family of molecular nanomagnets (MNM) with unsupported Ln-TM bonds and paves the way towards molecular intermetallics with strong direct magnetic exchange interactions—a promising route towards high-performance single-molecule magnets.

[1] Faculty of Chemistry, Jagiellonian University, Gronostajowa 2, 30-387 Kraków, Poland. [2] Marian Smoluchowski Institute of Physics, Jagiellonian University, Łojasiewicza 11, 30-348 Kraków, Poland. [3] Maastricht Science Programme, Faculty of Science and Engineering, Maastricht University, Paul-Henri Spaaklaan 1, 6229 EN Maastricht, The Netherlands. ✉email: dawid.pinkowicz@uj.edu.pl

Rare-earth intermetallic magnets—samarium-cobalt (SmCo)[1] and neodymium (NdFeB)[2,3]—are the strongest permanent magnets known to date with multiple applications in modern technology ranging from hard disk drives to electric vehicles and wind turbines[4]. Their commercial success results from the combination of the strong magnetic anisotropy of rare-earth ions and their direct magnetic coupling with transition metal ions. At the other far end of the current search for novel magnetic materials are magnetic molecules, called molecular nanomagnets[5] (MNMs) or single-molecule magnets (SMMs). MNMs are molecule-sized objects with magnetic memory effects governed by quantum mechanics[6–10]. They are strongly believed to revolutionize magnetic information storage, but at the moment do not have any real-world applications, as their performance is limited by low blocking temperatures and quantum tunneling of magnetization (QTM)[11]. Current trends in the design of MNMs[12,13] were ignited by the discovery of terbium double-decker [TbPc$_2$]$^-$ (Pc = phthalocyanine dianion)[14] and were fueled by the seminal perspective of Rinehart and Long[15]. Design principles focus on the precise control of the coordination sphere of a single lanthanide center to maximize the easy-axis magnetic anisotropy of the complex and to limit the spin-lattice relaxation rates[16–19]. While this approach led to several breakthroughs, including the observation of the magnetic hysteresis loop of molecular origin above the liquid nitrogen barrier for dysprosocenium[6,7] and magnetic field control of the magnetization blocking barrier[20], it appears to be reaching its limit as the proposed modifications were not as successful as expected[21,22].

The original approach to MNMs was based on designing multinuclear systems with strong magnetic superexchange, where several paramagnetic centers are glued together by bridging ligands to form a high-spin molecule. If magnetic anisotropy is present in such a system, it shows MNM properties[23,24]. An MNM of this type stands behind the success story of the whole field as it was indeed started by the discovery of a dodecanuclear carboxylate-bridged cluster Mn$_{12}$—a high-spin molecule ($S = 10$) comprising eight Mn$^{III}$ and four Mn$^{IV}$ ions interacting magnetically via the bridging carboxylate ligands[25–27]. However, the indirect character of these magnetic interactions, based on a weak superexchange coupling mechanism, limits the performance of Mn$_{12}$ to very low temperatures, below the boiling point of liquid helium. Attempts to adapt this particular strategy to dysprosocenium complexes suffer from a similar problem –weak superexchange interactions of the lanthanides with other metal centers through diamagnetic bridging ligands limit their influence on the slow magnetic relaxation[28–30]. Therefore, the route towards high-performance MNMs based on superexchange interactions has been largely abandoned.

Overall, despite the huge progress in the field initiated by Mn$_{12}$, its ultimate goal—the room temperature molecular nanomagnet (RT-MNM)—remains elusive. However, the aforementioned examples of the rare-earth intermetallics: SmCo$_5$[1] and Nd$_2$Fe$_{14}$B[2] provide clues for the possible direction in the design of RT-MNMs. Focusing the efforts on molecules comprising a rare-earth metal center coordinated solely by transition metal ions[31–34], mimicking the first coordination sphere of the Sm center in the SmCo$_5$ magnet (Fig. 1A–C) could enable the appearance of the direct exchange coupling between the highly anisotropic lanthanide central ion and the coordinated transition metals. This approach was recently predicted to efficiently suppress QTM in MNMs[35] and demonstrated for the ultra-hard MNM (CpiPr$_5$)$_2$Dy$_2$I$_3$ where the direct exchange is operational[36]. Interestingly, even much weaker indirect interactions were reported to be quite efficient in this matter[37].

The concept of unsupported bonds between the lanthanide and the transition metal was first introduced and explored by

Kempe et al.[32,34]. Later, it was proposed by Rinehart and Long[15] as a possible strategy for molecular nanomagnets and put to use by Nippe et al. reporting MNMs with unsupported direct bonds between the dysprosium ion and $4d$ (Fe) or $4d$ (Ru) transition metal ions[38]. However, the magnetic memory effect (magnetic hysteresis) has not been observed, most probably due to the unfavorable ligand field geometry[38] or the unfortunate choice of the rare-earth metal[32]. The introduction of $p$-block heavy metals directly into the coordination sphere of the lanthanide was also pursued resulting in interesting examples of MNMs[39,40]. Other systems involving direct bonds between actinides and $d$-metal elements were also reported, but slow magnetization dynamics were not studied[41–43]. Noteworthy, the concept of placing transition metal ions in the proximity of Ln-based MNMs employing suitable bridging ligands was pursued for tens of years with several examples showing enhancement of the slow relaxation of the magnetization due to the presence of the transition metal[44–46]. However, this strategy, while providing very good control of the connectivity of the target multi-metallic compounds, cannot warrant strong coupling between the two metal ions due to the weak overlap of the relevant orbitals of the lanthanide and the bridging moiety. Only the direct bonding of the transition metal to the lanthanide center could result in strong electronic and magnetic interactions.

Here, we apply the principles of easy-axis magnetic anisotropy design with the methodology of Kempe et al. that led previously to rare-earth metal complexes coordinated solely by transition metals[32]. We demonstrate an important step toward molecular intermetallic nanomagnets—a perfectly trigonal planar [Er$^{III}$(Re$^{I}$Cp$_2$)$_3$] complex (**ErRe$_3$**; Cp = cyclopentadienyl anion) with magnetic hysteresis up to 7.2 K (at 2.2 mT s$^{-1}$ magnetic sweep rate) where Er$^{III}$ is solely coordinated by three diamagnetic Re$^{I}$ ions.

## Results and discussion

**Synthetic strategy**. **ErRe$_3$** is obtained following the literature procedure for [Sm$^{III}$(Re$^{I}$Cp$_2$)$_3$][32]. The starting material [Er$^{III}$(btmsm)$_3$] (obtained according to literature procedures[47,48]; btmsm = bis(trimethylsilyl)methyl anion) reacts with [Cp$_2$ReH][49] in benzene (Fig. 1D). The btmsm$^-$ ligand coordinated to Er$^{III}$ acts as a strong base capable of deprotonating [Cp$_2$ReH]—a weak acid. Deprotonation of [Cp$_2$ReH] leads to the elimination of the byproduct bis(trimethylsilyl)methane ("alkane elimination"[34]) and the formation of the anionic [Cp$_2$Re$^{I}$]$^-$ species that readily coordinate to Er$^{III}$ resulting in the formation of a trigonal planar complex [Er$^{III}$(Re$^{I}$Cp$_2$)$_3$].

**Crystal structure**. **ErRe$_3$** crystallizes slowly from benzene in the form of small yellow prism crystals which were characterized structurally by single-crystal X-ray diffraction (SCXRD; trigonal $R$-3; Supplementary Table 1). The SCXRD structural analysis confirmed the coordination of three Re$^{I}$ ions to the Er$^{III}$ center with the formation of a nearly perfect trigonal planar neutral complex [Er$^{III}$(Re$^{I}$Cp$_2$)$_3$] (Fig. 1C) with the Re-Er-Re angles very close to 120° (121.22(2), 119.15(2), and 119.62(2)°) and the Er atom lying only 0.013(1) Å above the Re$_3$ plane. The cyclopentadienyl ligands are coordinated solely to Re$^{I}$ and arranged in a slightly tilted manner below and above the equatorial plane of the **ErRe$_3$** molecule (Fig. 1C). There are no other coordination bonds to Er$^{III}$. Other intramolecular contacts of Er$^{III}$ involve Cp ligands with the shortest Er$\cdots$C distances in the 2.829(8)–3.056(13) Å range—well beyond typical coordination bonds of lanthanide complexes. The Er-Re distances in **ErRe$_3$** (2.9004(5), 2.9124(5), and 2.9172(5) Å) are similar to those reported for [Sm$^{III}$(Re$^{I}$Cp$_2$)$_3$], [La$^{III}$(Re$^{I}$Cp$_2$)$_3$], and [Lu$^{III}$(Re$^{I}$Cp$_2$)$_3$][32] as well as other molecular compounds with unsupported rare earth—transition metal bonds[34,38] and the

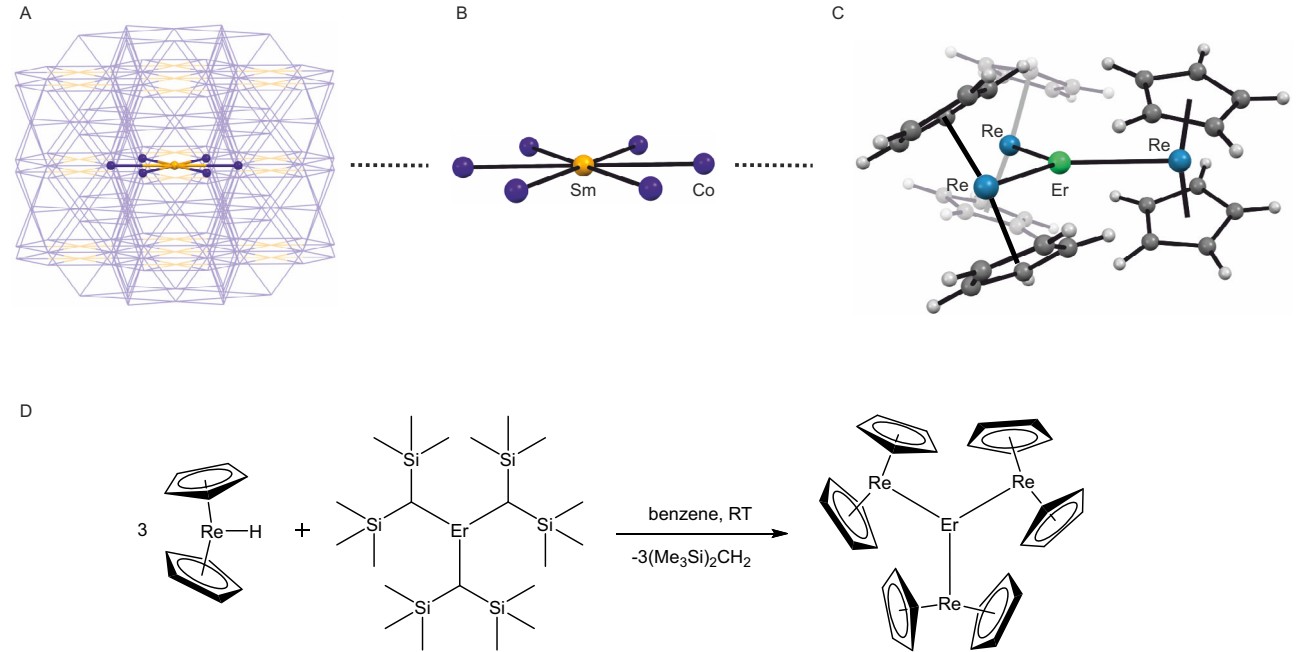

**Fig. 1 The synthetic strategy towards ErRe3 and its structure.** Graphical representation of the structural design transfer from the intermetallic magnet SmCo$_5$ (**A**, **B**) to the molecular nanomagnet [Er$^{III}$(Re$^I$Cp$_2$)$_3$] (**C**). The hexagonal planar SmCo$_6$ coordination with unsupported coordination bonds between the center of the rare earth and six transition metals in SmCo$_5$ (**A**, **B**) is mimicked by the trigonal planar coordination in **ErRe$_3$** (**C**). All three panels **A**–**C** are based on single-crystal structural models of SmCo$_5$[1] and **ErRe$_3$**. Sm-Co bonds in SmCo$_5$: 2.888 Å; Er-Re bonds in **ErRe$_3$**: 2.9004(5), 2.9124(5), 2.9172(5) Å. Panel **D** presents the synthesis of **ErRe$_3$** based on the work of Kempe et al.[32]. This approach utilizes the reaction of the weak Brønsted acid [Cp$_2$ReH] with the strong Brønsted base btmsm$^-$ in [Er$^{III}$(btmsm)$_3$] in benzene at room temperature (RT) resulting in (Me$_3$Si)$_2$CH$_2$ elimination and the formation of [Er$^{III}$(Re$^I$Cp$_2$)$_3$].

intermetallic SmCo$_5$ with Sm-Co distances of 2.888 Å[1]. The comparison of the Re-Cp distances in **ErRe$_3$** and the [Cp$_2$ReH] starting material (SCXRD structural analysis of [Cp$_2$ReH] was performed as part of this study Supplementary Table 1; CCDC 2027573) confirms the change of the valence state of the Re centers upon coordination to Er$^{III}$. The slightly unsymmetrical and tilted [Cp$_2$ReH] with the average Re-Cp$_{centroid}$ distance of 1.877(44) Å is similar to other compounds of this type[50–52] and shows slight shortening of these contacts in **ErRe$_3$** to av. 1.864(8) Å (Supplementary Fig. 1). The [Er$^{III}$(Re$^I$Cp$_2$)$_3$] molecules are stacked on top of each other in a hexagonal fashion along the $c$ crystallographic direction, leading to the formation of channels filled with disordered benzene molecules (Supplementary Fig. 2). The shortest intermolecular distances between the Er$^{III}$ centers can be found within the aforementioned stacks and amount to 7.6318(6) Å. There are six more nearest Er$^{III}$ neighbors within the 10 Å radius with distances larger than 9 Å (Supplementary Fig. 3). This ensures magnetic isolation of the individual [Er$^{III}$(Re$^I$Cp$_2$)$_3$] molecules that is sufficient for the observation of slow magnetization dynamics. Before attempting the magnetic characterization **ErRe$_3$** was subjected to a rigorous purity verification by performing a powder X-ray diffraction (PXRD) experiment for a sample loaded into a 0.3 mm glass capillary immersed in benzene and sealed using high vacuum grease (similar conditions were applied during the magnetic measurements discussed below). The experimental PXRD pattern shows very narrow peaks and matches almost perfectly the simulated one using 270 K SCXRD data (Supplementary Fig. 4).

**Magnetic properties and calculations.** The unusual coordination environment of Er$^{III}$ in **ErRe$_3$** may cast doubts on the nature of its electronic ground state. However, magnetic measurements for **ErRe$_3$** confirm the valence states of the central Er$^{III}$ ion and the diamagnetic Re$^I$ donor atoms. The $\chi T$ ($\chi$—molar magnetic susceptibility) value of 11.3 cm$^3$ K mol$^{-1}$ at 260 K is close to 11.48 cm$^3$ K mol$^{-1}$ expected for a single Er$^{III}$ $^4$I$_{15/2}$ ground multiplet (Supplementary Fig. 5). A similar agreement between the experiment and the expected $\chi T$ was reported for [Sm$^{III}$(Re$^I$Cp$_2$)$_3$] ($^6$H$_{5/2}$ multiplet)[32]. This is further confirmed by a very good agreement of the experimental $\chi T$ and the ab initio calculations (red solid line in Supplementary Fig. 5 and supplemental information) using Molcas[53,54]. The $\chi T(T)$ dependence shows an abrupt increase below 18 K, which can be ascribed to ferromagnetic intermolecular interactions and blocking of magnetization.

The $M(H)$ ($M$—molar magnetization, $H$—magnetic field) measurements (magnetic field sweep rate of 2.2 mT s$^{-1}$) revealed the presence of waist-restricted (pinched) magnetic hysteresis loops up to 7.2 K (Fig. 2A), originating from the slow relaxation of the magnetization of the trigonal **ErRe$_3$**. This temperature matches Er$^{III}$ sandwich and metallocene complexes[55]. Furthermore, the hysteresis loop in **ErRe$_3$** opens in the 0.05–1.5 T range at 1.8 K and is the widest among trigonal Er$^{III}$ MNMs.

The slow magnetization dynamics were studied and confirmed by the alternating current (AC) magnetic susceptibility measurements performed in the 1–10000 Hz frequency range up to 27 K in the absence of the external magnetic field (Fig. 2C and Supplementary Table 2). However, the resulting temperature dependence of the magnetic relaxation rate $\tau^{-1}(T)$ (Fig. 2D) contains contributions from three processes: quantum tunneling of magnetization (QTM), Raman and Orbach described by five parameters included in Eq. 1:

$$\tau^{-1} = A + CT^n + \tau_0^{-1}\exp\left(\frac{-U_{eff}}{k_B T}\right) \qquad (1)$$

hence, all relevant parameters: $A$, $C$, $n$, $\tau_0$, and, most importantly, the effective energy barrier for the magnetization reversal $U_{eff}$ arising from the Orbach relaxation, can be

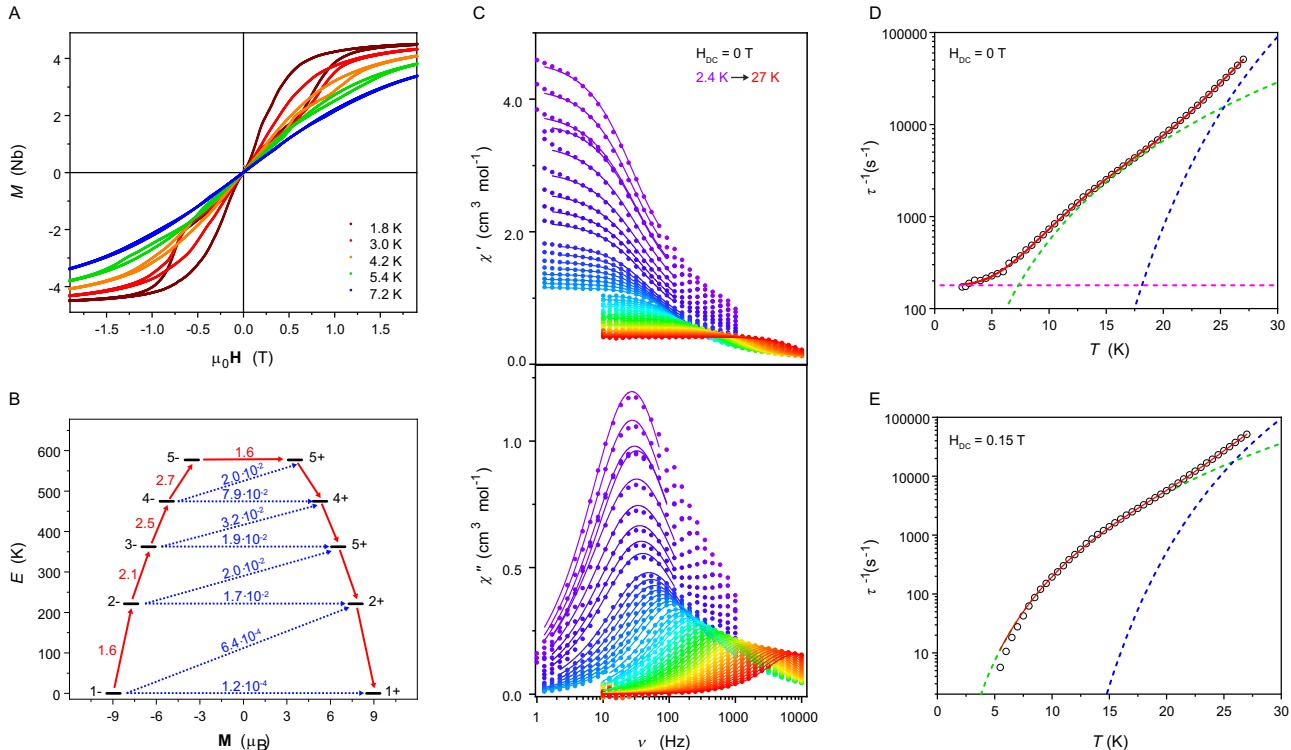

**Fig. 2 Magnetic properties of ErRe₃.** Magnetization ($M$) hysteresis loops were recorded with the 2.2 mT s⁻¹ magnetic field ($\mu_0\mathbf{H}$) sweep rate in the −2 to 2 T range (**A**). The energy of the calculated Kramers Doublets (KDs) arising from the splitting of the $^4I_{15/2}$ multiplet of $Er^{III}$ in **ErRe₃** with the most probable relaxation pathway represented by red arrows (**B**). In-phase ($\chi'$) and out-of-phase ($\chi''$) AC molar magnetic susceptibility recorded at zero magnetic field in the 2.4–27 K temperature range, demonstrating the slow magnetization dynamics of **ErRe₃**, solid lines are the best fits to the generalized Debye model with parameters presented in Supplementary Table 2 (**C**). Temperature dependence of the magnetization relaxation rate $\tau^{-1}(T)$ at zero (**D**) and 0.15 mT (**E**) magnetic field obtained from the generalized Debye fitting of the respective AC molar magnetic susceptibility measurements (circles—experimental points, red line—best fits using Eq. 1 (**D**) and Eq. 2 (**E**), blue dashed line—Orbach relaxation, green dashed line—Raman relaxation, magenta dashed line—QTM relaxation). Standard deviations in **A**, **C**, **D**, and **E** are smaller than the size of the data points.

extracted only with limited confidence from AC data collected at zero DC field (best-fit parameters $A = 179(4)$ s⁻¹, $C = 0.137(9)$ K⁻ⁿ s⁻¹, $n = 3.60(2)$, $\tau_0 = 8.3(8)\cdot10^{-10}$ s and $U_{eff} = 285(3)$ K with $R^2 = 0.99994$; Fig. 2D). To mitigate this issue, the AC magnetic studies were repeated under an applied DC magnetic field of 0.15 T (Supplementary Fig. 6 and Supplementary Table 3). The presence of a small applied magnetic field quenches the QTM with only a negligible direct process, which was accurately determined from the magnetic field dependence of the relaxation time (Supplementary Fig. 7 and Supplementary Table 4). Thus, the temperature dependence of the relaxation rate under $H_{DC} = 0.15$ T could be reliably fitted including only the Raman and Orbach relaxation mechanisms using Eq. 2:

$$\tau^{-1} = CT^n + \tau_0^{-1}\exp\left(\frac{-U_{eff}}{k_B T}\right) \quad (2)$$

where $\tau^{-1}$ is the magnetic relaxation rate, $C$ and $n$ parameters describe the Raman-like process, and $\tau_0$ and $U_{eff}$ are related to the Orbach process. The satisfactory fit could be obtained with $C = 0.0035(4)$ K⁻ⁿ s⁻¹, $n = 4.74(3)$, $\tau_0 = 3.0(5)\cdot10^{-10}$ s, and $U_{eff} = 314(5)$ K with $R^2 = 0.99987$ (Fig. 2E). The relation between $U_{eff}$ and $\tau_0$ falls perfectly in the statistically expected range for prolate Ln MNMs as analyzed by Gaita-Ariño et al. in their work covering 1405 examples of SIMs[12]. The observed low Raman exponent of 4.74 is most probably caused by spin-lattice coupling with optical phonons as summarized and discussed by Gu and Wu[56].

The obtained effective energy barrier for the magnetization reversal $U_{eff} = 314(5)$ K is significantly smaller than expected from the theoretical calculations (CASSCF-RASSI; OpenMolcas 19.11[53]; for details see Supplementary Information including Supplementary Tables 5 and 6 with the information on basis sets, spin-orbit energies and g-tensors of the lowest Kramers Doublets, respectively), predicting the fastest relaxation between the 5± Kramers Doublet lying 576 K above the highly axial ground state (Fig. 2B and Supplementary Table 6). The value 314(5) K is therefore underestimated. Additional fittings of the Raman and Orbach processes with fixed $U_{eff}$ values corresponding to the calculated energies of the lower KDs were performed: 2± of 224 K, 3± of 363 K, and 4± of 475 K. The best fits (Supplementary Fig. 8A–C) are characterized by the following $R^2$ values: 0.99702 for $U_{eff\,2\pm} = 224$ K, 0.99960 for $U_{eff\,3\pm} = 363$ K, and 0.99808 for $U_{eff\,4\pm} = 475$ K. The best agreement is observed for $U_{eff\,3\pm} = 363$ K, which corresponds to the energy of the third KD. This is also closest to the value obtained from the free fitting of the relaxation processes presented in Fig. 2E. An attempt to describe the magnetic relaxation of **ErRe₃** using only the Raman-like process leads to a very poor fit as depicted in Supplementary Fig. 8D. Thus we conclude, that the observed Orbach relaxation for **ErRe₃** proceeds predominantly through the third Kramers doublet, characterized by the estimated energy barrier $U_{eff}$ approaching 363 K. This value is three times higher than that observed for the best trigonal[48,57] or low-coordinate[58,59] $Er^{III}$ molecules and is comparable to the $Er^{III}$ sandwich and metallocene complexes[37,55,60–64]. Interestingly, **ErRe₃** shows significantly slower relaxation as compared to [Er(N(SiMe₃)₂)₃] characterized by $U_{eff} = 122$ K[57] and surpasses the theoretical limit of 306 K (212.6 cm⁻¹) for any trigonal planar molecule

based on the theoretical analysis of a hypothetical $Er^{III}(NH_2)_3$ compound[65]. **ErRe₃** is a clear demonstration that direct bonding of transition metals to lanthanides might be one of the most promising directions towards room temperature MNMs, especially if paramagnetic donor atoms could be used.

Our findings regarding the magnetic dynamics of **ErRe₃** are consistent with the recent data analysis by Gaita-Ariño et al[12]. The analysis shows relatively high magnetic hysteresis temperatures for prolate systems such as equatorial $Er^{III}$ SIMs (coordinated by rigid ligands) accompanied by the maxima of the out-of-phase magnetic susceptibility ($\chi''$) at relatively low temperatures. **ErRe₃** is a perfect demonstration of this trend with the $T_{hyst}$ of 7.2 K and $U_{eff} = 314$ K.

In conclusion, the reported triangular molecule $[Er^{III}(Re^ICp_2)_3]$ constitutes a molecular nanomagnet that mimics structurally the hardest commercial rare-earth magnet known $SmCo_5$ and creates a connection between two incompatible fields: the rare-earth intermetallics and molecule-based compounds. **ErRe₃** is a rare example of a lanthanide single-molecule magnet coordinated solely by transition metals which enables the formulation of a promising strategy towards molecular intermetallic nanomagnets, where the lanthanide—the source of strong magnetic anisotropy—is directly coupled to the spins of softer transition metal ions playing the role of ligands.

## Methods

**General considerations**. All manipulations were performed under an argon gas atmosphere inside the Inert PureLab HE glovebox ($O_2 < 0.1$ ppm and $H_2O < 0.5$ ppm). All solvents used in this study (except benzene) were of HPLC grade and were additionally dried under argon using the Inert PureSolv EN7 solvent purification system and stored over 3 Å molecular sieves for at least 24 h before use. Anhydrous benzene was purchased from commercial sources (Merck) and degassed by performing three freeze-pump-thaw cycles before use. Anhydrous $ReCl_5$ (at least 99.9%) was purchased from Merck and used as received for the preparation of $[Cp_2ReH]$ according to the modified literature procedure[49] described below. The obtained $[Cp_2ReH]$ was characterized by means of SCXRD and the purity was confirmed by performing PXRD measurements. $[Er^{III}(btmsm)_3]$ was also prepared following the modified literature procedures[47,48]—see below for the description of the preparation procedure. The purity of the obtained product was checked by PXRD. See Supplementary Information file for measurement details and ab initio calculations. Warning! Benzene is recognized as carcinogenic to humans and should be handled with caution using a well-ventilated hood or glovebox.

**Preparation of $[Er^{III}(btmsm)_3]$**. $[Er^{III}(btmsm)_3]$ was prepared following the modified literature procedures[47,48]. Inside the Ar-filled glovebox bis(trimethylsilyl) methyllithium[66] (Libtmsm; 0.91 g, 5.47 mmol) was slowly added in small portions to a stirred clear pink solution of $[Er^{III}(BHT)_3]$[48] (1.13 g, 1.37 mmol; BHT = 2,6-Di-tert-butyl-4-methylphenolate) in 17 ml of pentane. After the addition is finished the reaction mixture is still slightly turbid due to the undissolved Libtmsm, but becomes clear after ca. 20 min of stirring and then turbid again after ca. 1 h of stirring due to the precipitation of the byproduct LiBHT. The stirring is continued for 5 h at room temperature and then the thick suspension is vacuum filtered through a P4 fritted glass funnel in order to remove LiBHT. Please note that the funnel needs to be thoroughly clean, rinsed with THF, and dried at 140 °C prior to use. The product might start to crystallize in the filtrate during vacuum filtration. The pink filtrate is transferred into a tightly sealed glass vial and kept at −40 °C for 5 h for crystallization. The obtained pink needle crystals are filtered by vacuum filtration using a PTFE membrane, collected, and stored in the freezer. Yield: 350 mg (39%). Additional crop of the product (100 mg; 10%) is obtained by washing the LiBHT byproduct collected after the reaction with ca. 10 ml of pentane and keeping the washings at −40 °C for 24 h. The purity of both crops was confirmed by collecting PXRD patterns, which were identical to that simulated from the scXRD data.

**Preparation of $[Cp_2ReH]$**. $[Cp_2ReH]$ was synthesized according to the modified literature procedure[49]. Inside the Ar-filled glovebox, a 100 mL 29/32 Schlenk flask was charged with a large magnetic stirrer, sodium cyclopentadienide (NaCp; 3.55 g, 0.04 mol), and 25 mL THF, and was cooled to −40 °C. The cold mixture was set on a fast stirring and $ReCl_5$ (2.18 g, 0.006 mol) was added in small portions. The resulting mixture was left to heat to room temperature, then NaBH₄ (0.57 g, 0.015 mol) was added, the flask was sealed with a glass stopper, and was removed from the glovebox. Using standard Schlenk line techniques, the mixture was refluxed under Ar for 2.5 h, then it was cooled to room temperature and the solvent was removed under vacuum. The resulting brown solid was dried under vacuum for 30 min at 70 °C, then the reaction flask was fitted with a 29/32 cold finger filled with dry ice/acetone mixture and the brown solid was sublimed under vacuum. The temperature during the sublimation was gradually increased up to 180 °C under vacuum ($p \approx 10^{-2}$ mbar) and continued until the product was deposited on the cold finger surface as a brownish-yellow powder, leaving most of the brown residue on the bottom of the reaction flask. The reaction flask was moved inside the glovebox, the yellow product was moved to another Schlenk flask and the vacuum sublimation was repeated at 120 °C (this time with ice-cooling of the cold finger) yielding $Cp_2ReH$ as yellow needles. Yield: 0.50 g (26%). The purity of the product was confirmed by collecting the PXRD pattern, which was identical to that simulated from the scXRD data.

**Preparation of $[Er^{III}(Re^ICp_2)_3]\cdot0.5C_6H_6$ (ErRe₃)**. All manipulations were performed in an argon-filled glovebox (Inert PureLab HE). $[Cp_2ReH]$ (242 mg, 0.76 mmol) was dissolved in anhydrous benzene (3.5 ml) and added to the benzene solution (2.5 ml) of $[Er^{III}(btmsm)_3]$ (160 mg, 0.25 mmol). The resulting orange solution was stirred using a glass rod for 5 min and then stored in a 20 ml scintillation vial that was left open for 5 h inside the glovebox. After this time yellow-orange crystals appeared which were collected by vacuum filtration using a 1-µm Teflon membrane (Whatman). Yield: 70 mg (0.061 mmol; 24%). The purity of the compound was checked by powder X-ray diffraction, with the experimental pattern (Supplementary Fig. 4) matching perfectly the simulated one from the SCXRD structural model obtained at 270 K (CCDC 2065531). IR, neat (cm$^{-1}$): 796 s, 808 s, 816 s, 832 m, 850 m, 987 s, 1000 s, 1059 s, 1087 s, 1095vs, 1154vw, 1188vw, 1201vw, 1255 m, 1273 m, 1342 m, 1358 m, 1394 s, 1419 s, 1462 m, 1479 m, 1558 s, 1657s, 2001vw, 2034vw, 2854 s, 2929vs, 3036 m, 3063 s, 3089 s, 3550b (Supplementary Fig. 9).

## Data availability

The data that support the findings of this study can be found in the manuscript or the accompanying supplementary information files. Source data for all Figures are provided with this paper as a single Excel file and are also available from the corresponding author: Dawid Pinkowicz (dawid.pinkowicz@uj.edu.pl). Crystallographic data generated during this study were deposited with the Cambridge Structural Database and can be obtained free of charge from the Cambridge Crystallographic Data Centre via www.ccdc.cam.ac.uk/data_request/cif (**ErRe₃** at 100 K: CCDC 2065530, **ErRe₃** at 270 K: CCDC 2065531 and $[Cp_2ReH]$ at 100 K: CCDC 2027573). Source data are provided with this paper.

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

## Acknowledgements

This work was financed by the Polish National Science Centre within the Sonata Bis 6 project no. 2016/22/E/ST5/00055 (D.P.). The research was partly carried out with the equipment purchased thanks to the financial support of the European Regional Development Fund in the framework of the Smart Growth Operational Program 2014-2020, contract no. POIR.04.02.00-00-D001/20-00 (D.P.). The open-access publication of this article was funded by the program "Excellence Initiative – Research University" at the Jagiellonian University in Krakow. The authors acknowledge Alejandro Gaita-Ariño for

pointing out and formulating an important observation regarding the slow magnetic dynamics of the reported compound **ErRe₃** in comparison to known Ln-SIMs.

## Author contributions

M.M. prepared the starting material [Cp$_2$ReH], participated in the preparation of the target compound, performed and analyzed the magnetic measurements, and wrote the relevant fragment of the manuscript; M.B. prepared the starting material [Er$^{III}$(btmsm)], participated in the preparation of the target compound, analyzed its structure and wrote a relevant fragment of the manuscript; S.B. participated in the AC magnetic measurements and their analysis. V.V. performed and analyzed the ab initio calculations and wrote the relevant fragment of the manuscript. D.P. acquired funding, designed the experiments, performed the synthesis of the target compound, coordinated and supervised the research, and wrote the manuscript. All authors reviewed and agreed to the final version of the paper.

## Competing interests

The authors declare no competing interests.
