## [Peer Review File · Nature Communications]

An Intermetallic Molecular Nanomagnet with the Lanthanide Coordinated Only by Transition MetalsREVIEWER COMMENTS

Reviewer #1 (Remarks to the Author):

The authors combine a known strategy, namely equatorial Er³⁺ coordination with relatively rigid ligands, with an exciting and novel approach, namely direct coordination of lanthanide ions by transition metals, and report a significant and interesting molecular nanomagnet. This work presents magnetically important, chemically novel results, which are also unusually timely after the ultra-hard molecular nanomagnet breakthrough recently reported in Science. I recommend it to be published as-is, or, at most, with minor modifications.

I include here a few minor comments:

-As a minor readability note, to facilitate understanding for non-experts, one could explicitly state "diamagnetic" when introducing the Re(I)-based ligand, rather than just implicitly suggesting its diamagnetic nature e.g. by confirming that the molar XT value at 260K for the complex is close to what would be expected for a single Er(III) complex.

-To avoid confusions for synthetic chemists looking for inspiration, it also might be good to briefly clarify the similarities and differences between the current and the (extensively explored) approach of placing a diamagnetic transition metal in the proximity of the Ln ion.

-A minor positive point to put the current results in perspective with known data: the experimental findings reported herein in terms of slow vs fast magnetic dynamics are consistent with recent data analysis showing relatively high hysteresis temperatures for equatorial Er³⁺ coordination with rigid ligands, compared with relatively low maxima of out-of-phase susceptibility for the same systems; in contrast, for Tb and Dy SIMs, hysteresis in the 6-10K range would correspond to higher ac maxima (at typical 1000Hz) in a 30-60K range. The relation between U_{eff} and tau₀ in their fit is also in the statistically expected range for Er³⁺ SIMs (and lower values for tau₀*U_{eff}^{2.4} than what would have been expected for Dy³⁺ or Tb³⁺ SIMs). [<https://www.researchsquare.com/article/rs-490959/v1> , <https://go.uv.es/rosaleny/SIMDAVIS>] I understand these coincidences with the existing dataset as an independent validation of the results.

-While I would not recommend to do this in the current work, it would be interesting, in the future, to substantially extend the active space considered in the OpenMolcas studies to include the d orbitals of the Re(I) ions. This may give a more profound insight into what makes this system special.

-I don't think it's really consistent to claim that the ~7Å Er-Er separation means a sufficiently good magnetic isolation (which implicitly assumes that no magnetic moment is delocalized towards the Re ions) and at the same time justify the anomaly in Fig S5 as resulting from a ferromagnetic interaction between neighbouring molecules.

Alejandro Gaita-Ariño

Reviewer #2 (Remarks to the Author):

This paper describes the synthesis and magnetic properties of an erbium complex with three [Re|Cp₂]- ligands, with Er-Re bonds. The work seems well done, and the compounds have been perfectly characterised. However, I believe that the work does not merit publication in Nature Commun. This is because the authors tell us that this would be a good alternative to try to improve the magnetic properties of current SMMs, whose record blocking temperature is 80 K. But this is not reflected at all in the magnetic data: the blocking temperature of this compound is less than 2 K, with a zero coercive magnetic field. Moreover, the compound is very unstable in air and, to make matters worse, it is prepared in benzene, a solvent which has been shown to be carcinogenic. I do not

therefore see what is new about the work of Layfield et al (their reference 7) or Chilton et al (their reference 36). Accordingly, I believe that the work is not original enough to be published in Nat. Commun., and should therefore be rejected.

Reviewer #3 (Remarks to the Author):

This is a high quality paper in terms of data presented and the manner in which it is conveyed. The paper is well written and concise. The figures are clear. The topic is also of significant current interest. That being said, my initial thought was that because of ref. 34, the paper should be published in a more specialized journal like IC. However, upon reading ref. 34 carefully, it is clear that the authors of this study are being self-effacing. The Er-Re₃ core that they are presenting might have been derived from a similar synthetic methodology used to prepare Ln-TM dinuclear cores in ref. 34, but that is where the similarities end. This molecule is distinct and displays single molecule magnetism that will motivate others to pursue this route for mimicking the core of SmCo₅. In short, I think it should be accepted by Nat. Commun.

Response to the reviewers

Reviewer #1: Alejandro Gaita-Ariño

Thank you very much for the positive feedback on our work. We have modified our manuscript according to your comments (please see below):

1. As a minor readability note, to facilitate understanding for non-experts, one could explicitly state "diamagnetic" when introducing the Re(I)-based ligand, rather than just implicitly suggesting its diamagnetic nature e.g. by confirming that the molar χT value at 260K for the complex is close to what would be expected for a single Er(III) complex

We have added an explicit statement regarding the diamagnetism of the coordinated Cp₂Re-ligand as suggest by the reviewer.

2. To avoid confusions for synthetic chemists looking for inspiration, it also might be good to briefly clarify the similarities and differences between the current and the (extensively explored) approach of placing a diamagnetic transition metal in the proximity of the Ln ion.

The extensively explored approach towards bimetallic complexes combining transition metals (TM) and lanthanides (Ln) bridged by various ligands (L) is well-established and produced numerous TM-L-Ln compounds where the TM is in a close vicinity of the Ln ion (Yan, D., Joana, C., Lorena, R., Salvador, C., José, J.B., and Alejandro, G.-A. (2021). Data mining, dashboards and statistics: a powerful framework for the chemical design of molecular nanomagnets. Research Square. 10.21203/rs.3.rs-490959/v1. While this approach can enhance the energy barrier for the magnetization reversal, the bridging ligands provide only weak or no electronic coupling between the two metal ions. In the approach demonstrated in our study, the direct TM-Ln bonds may lead to strong or even very strong interactions resulting in unprecedented electronic and magnetic properties. Chemically, the proposed approach is significantly more challenging as there are no universal strategies towards direct TM-Ln bonded compounds as compared to TM-L-Ln complexes which can be easily designed using coordination chemistry and crystal engineering methods. A few relevant sentences have been added in the introduction section together with references 44-46:

"Noteworthy, the concept of placing transition metal ions in the proximity of Ln-based MNMs by means of suitable bridging ligands was pursued for tens of years with several examples showing enhancement of the slow relaxation of the magnetization due to the presence of the transition metal.^{44, 45, 46} However, this strategy, while providing very good control of the connectivity of the target multi-metallic compound, cannot warrant strong coupling between the two metal ions due to the weak overlap of the relevant orbitals of the lanthanide and the bridging moiety. Only the direct bonding of the transition metal to the lanthanide center could result in strong electronic and magnetic interactions."

*3. A minor positive point to put the current results in perspective with known data: the experimental findings reported herein in terms of slow vs fast magnetic dynamics are consistent with recent data analysis showing relatively high hysteresis temperatures for equatorial Er³⁺ coordination with rigid ligands, compared with relatively low maxima of out-of-phase susceptibility for the same systems; in contrast, for Tb and Dy SIMs, hysteresis in the 6-10K range would correspond to higher ac maxima (at typical 1000Hz) in a 30-60K range. The relation between U_{eff} and τ_0 in their fit is also in the statistically expected range for Er³⁺ SIMs (and lower values for $\tau_0 * U_{\text{eff}}^{2.4}$ than what would have been expected for Dy³⁺ or Tb³⁺ SIMs). [<https://www.researchsquare.com/article/rs-490959/v1>, <https://go.uv.es/rosaleny/SIMDAVIS>] I understand these coincidences with the existing dataset as an independent validation of the results.*

Thank you for pointing it out, we've completely missed this important observation. We would like to add the following two sentences based entirely on this comment in the Results and Discussion section:

"The relation between U_{eff} and τ_0 falls perfectly in the statistically expected range for prolate Ln MNMs as analyzed by Gaita-Ariño et al. in their work covering 1405 examples of SIMs.¹²"

and

"Our findings regarding the magnetic dynamics of **ErRe₃** are consistent with the recent data analysis by Gaita-Ariño et al.¹² The analysis shows relatively high magnetic hysteresis temperatures for prolate systems such as equatorial Er^{III} SIMs (coordinated by rigid ligands) accompanied by relatively low maxima of the out-of-phase magnetic susceptibility (χ''). **ErRe₃** is a perfect demonstration of this trend with the T_{hyst} of 7.2 K and $U_{\text{eff}} = 314$ K."

We've also placed a sentence in the acknowledgments section referring to the reviewer's comment that was included in the manuscript:

"The authors acknowledge Alejandro Gaita-Ariño for pointing out and formulating an important observation regarding the slow magnetic dynamics of the reported compound **ErRe₃** in comparison to known Ln-SIMs."

4. While I would not recommend to do this in the current work, it would be interesting, in the future, to substantially extend the active space considered in the OpenMolcas studies to include the d orbitals of the Re(I) ions. This may give a more profound insight into what makes this system special.

Thank you for this suggestion. We are currently investigating other Re-Ln MNMs and we will definitely extend the active space in our calculations to include the *d* orbitals.

5. I don't think it's really consistent to claim that the ~7Å Er-Er separation means a sufficiently good magnetic isolation (which implicitly assumes that no magnetic moment is delocalized towards the Re ions) and at the same time justify the anomaly in Fig S5 as resulting from a ferromagnetic interaction between neighbouring molecules.

By "a sufficiently good magnetic isolation" we've meant separation that is sufficient for the observation of the slow magnetic relaxation of the molecular origin. However, we agree that this statement might be confusing. The relevant fragment of the text in the subsection describing the crystal structure of **ErRe₃** was modified:

"This ensures magnetic isolation of the individual [Er^{III}(Re^ICp₂)₃] molecules that is sufficient for the observation of slow magnetization dynamics."

Reviewer #2

Thank you very much for your critical review. We deeply disagree with your opinion.

However, I believe that the work does not merit publication in Nature Commun. This is because the authors tell us that this would be a good alternative to try to improve the magnetic properties of current SMMs, whose record blocking temperature is 80 K. But this is not reflected at all in the magnetic data: the blocking temperature of this compound is less than 2 K, with a zero coercive magnetic field.

- Indeed, the presented **ErRe₃** complex does not beat the Dy^{III} metallocene SIMs. It is, however, one of the best Er^{III} SIMs ever reported and this fact already validates the proposed strategy of using direct TM-Ln bonds in the design of high-performance molecular nanomagnets that mimic the structure of SmCo₅ and other permanent Ln-based magnets. Moreover, the recently published "ultra-hard molecular nanomagnet" (Science, 2022) also features metal-

metal bonding as the main responsible for its record breaking magnetic properties. As pointed out by reviewer #1, the presented results are "unusually timely" in this regard.

Moreover, the compound is very unstable in air...

- the compound is indeed unstable in air. However, the same applies to basically all organometallic Ln-SIMs including the Dy^{III} metallocenes (Guo et al. Science 2018, 362, 1400-1403 and Goodwin et al. Nature 2017, 548, 439-442) or the ultra-hard Dy₂ nanomagnet published this year (Gould et al. Science 2022, 375, 198-202).

...and to make matters worse, it is prepared in benzene, a solvent which has been shown to be carcinogenic.

- Organometallic SIMs are commonly synthesized using chemicals that are toxic or carcinogenic (Gould et al. J. Am. Chem. Soc. 2019, 141, 33, 12967–12973 or Goodwin et al. Nature 2017, 548, 439-442). If handled and disposed of responsibly, the risks are minimal. Nevertheless, we've added a suitable sentence in the experimental section in order to warn the readers that benzene causes cancer in humans and should be handled with great caution:
"Warning! Benzene is recognized as carcinogenic to humans and should be handled with caution using a well-ventilated hood or glovebox."

Reviewer #3

Thank you for the positive feedback on our manuscript.

REVIEWER COMMENTS

Reviewer #1 (Remarks to the Author):

With respect to my own comments, I am satisfied with the improvements. I would add that, for the interpretation of the fits of the ac data (Orbach-only vs Orbach+Raman vs energy level estimates via quantum chemistry calculations) the authors may benefit from considering the perspective of Gu, Wu, Phys. Rev. B (2021) 103, 014401. Gu and Wu performed a serious analysis of the detailed physical processes that may be underlying the general assumption of "Orbach+Raman" and give insights about how this is still a simplification.

With respect to the comments of Reviewer #2:

-I think issues with toxicity / stability (or lack thereof) are not relevant for publication of this work in Nature Communications, since the work is fundamental science rather than an attempt to reach mass production for applied purposes. However I do think that mentioning the issue improves the manuscript.

-I think the main criticism, while not enough to reject the work from Nature Communications, has some merits in terms of toning down the claims. The title is purely descriptive and not misleading, and the same can be said for the conclusions. However, the abstract, in its current wording, can be read as suggesting that the present work is aimed towards practical high-density data storage, or that the presented new family of compounds is about to beat the 80 K record for hysteresis. The manuscript would gain quality if the style here was toned down for realism.

I reiterate my recommendation of acceptance, perhaps after minor corrections.

Response to the Reviewer: Alejandro Gaita-Ariño

We have modified our manuscript according to the comments of the reviewer:

1. *With respect to my own comments, I am satisfied with the improvements. I would add that, for the interpretation of the fits of the ac data (Orbach-only vs Orbach+Raman vs energy level estimates via quantum chemistry calculations) the authors may benefit from considering the perspective of Gu, Wu, Phys. Rev. B (2021) 103, 014401. Gu and Wu performed a serious analysis of the detailed physical processes that may be underlying the general assumption of "Orbach+Raman" and give insights about how this is still a simplification.*

We have added a short sentence referring to the work of Gu and Wu with a short sentence explaining the low Raman exponent observed in the magnetic behavior of our compound:

"The observed low Raman exponent of 4.74 is most probably caused by spin-lattice coupling with optical phonons as summarized and discussed by Gu and Wu (ref. 56).

2. *I think issues with toxicity / stability (or lack thereof) are not relevant for publication of this work in Nature Communications, since the work is fundamental science rather than an attempt to reach mass production for applied purposes. However I do think that mentioning the issue improves the manuscript.*

We fully agree with this view.

3. *I think the main criticism, while not enough to reject the work from Nature Communications, has some merits in terms of toning down the claims. The title is purely descriptive and not misleading, and the same can be said for the conclusions. However, the abstract, in its current wording, can be read as suggesting that the present work is aimed towards practical high-density data storage, or that the presented new family of compounds is about to beat the 80 K record for hysteresis. The manuscript would gain quality if the style here was toned down for realism.*

We have modified the abstract by removing the sentences referring to high-density data storage and the hypothesis that the presented compound might beat the current 80 K threshold. The new abstract is included below and in the final manuscript:

"Inspired by the hexagonal structure of the hardest intermetallic magnet SmCo_5 , we have synthesized a nanomagnetic molecule where the central lanthanide (Ln) Er^{III} is coordinated solely by three transition metal ions (TM) in a perfectly trigonal planar fashion. This intermetallic molecule $[\text{Er}^{\text{III}}(\text{Re}^{\text{I}}\text{Cp}_2)_3]$ (**ErRe₃**) starts a family of molecular nanomagnets (MNM) with unsupported Ln-TM bonds and paves the way towards molecular intermetallics with strong direct magnetic exchange interactions – a promising route towards high-performance single-molecule magnets."